# *Trypanosoma brucei* Interaction with Host: Mechanism of VSG Release as Target for Drug Discovery for African Trypanosomiasis

**DOI:** 10.3390/ijms20061484

**Published:** 2019-03-25

**Authors:** Cláudia Jassica Gonçalves Moreno, Adriana Temporão, Taffarel Torres, Marcelo Sousa Silva

**Affiliations:** 1Programa de Pós-graduação em Bioquímica, Centro de Biociências, Universidade Federal do Rio Grande do Norte, Natal 59064-741, Brazil; claudia.mrn1@gmail.com; 2Instituto de Tecnologia Química e Biológica António Xavier, Universidade Nova de Lisboa, 2775-412 Oeiras, Portugal; adrianatemporao@gmail.com; 3Centro de Ciências Biológicas e da Saúde, Universidade Federal Rural de Semi-árido, Mossoró 59625-900, Brazil; taffarel.torres@ufersa.edu.br; 4Departamento de Análises Clínicas e Toxicológicas, Centro de Ciências da Saúde, Universidade Federal do Rio Grande do Norte, Natal 59012-570, Brazil; 5Global Health and Tropical Medicine, Instituto de Higiene e Medicina Tropical, Universidade Nova de Lisboa, 1349-008 Lisbon, Portugal

**Keywords:** *Trypanosoma brucei*, major surface protease, antigenic variation, variable surface glycoprotein, phospholipase-C

## Abstract

The protozoan *Trypanosoma brucei*, responsible for animal and human trypanosomiasis, has a family of major surface proteases (MSPs) and phospholipase-C (PLC), both involved in some mechanisms of virulence during mammalian infections. During parasitism in the mammalian host, this protozoan is exclusively extracellular and presents a robust mechanism of antigenic variation that allows the persistence of infection. There has been incredible progress in our understanding of how variable surface glycoproteins (VSGs) are organised and expressed, and how expression is switched, particularly through recombination. The objective of this manuscript is to create a reflection about the mechanisms of antigenic variation in *T. brucei*, more specifically, in the process of variable surface glycoprotein (VSG) release. We firstly explore the mechanism of VSG release as a potential pathway and target for the development of anti-*T. brucei* drugs.

## 1. Introduction

Human African trypanosomiasis, also known as sleeping sickness, is a vector-borne disease caused by two sub-species of protozoan parasites, *T. brucei rhodesiense* and *T. brucei gambiense* [1,2]. Both sub-species are morphologically indistinguishable but have considerable epidemiological and clinical differences [3].

The vectorial transmission of sleeping sickness occurs through the inoculation of metacyclic trypomastigote forms of *T. brucei* during the blood feeding of the tsetse fly (Glossina sp.) [4]. Metacyclic trypomastigote forms, also called slender forms, multiply through binary fission during parasitism in the mammalian host, mainly in blood, lymph, and cerebrospinal fluid [2].

During the life cycle, the slender forms evolve into the non-proliferative phase, also called stump forms, which are ingested by the tsetse fly during blood feeding in an infected vertebrate host. Subsequently, the parasites differentiate into the procyclic trypomastigote form in the midgut of the invertebrate vector [3,5]. Next, in the vector salivary glands, the parasites differentiate into metacyclic trypomastigotes. This stage is the infective form for the vertebrate host, characterized by the presence of variable surface glycoprotein (VSG), a dense layer of surface proteins that allows the survival and maintenance of the parasite inside the host by a mechanism of antigenic variation [6].

There are two clinical stages of sleeping sickness, the hemolymphatic and the neurological stages. Clinical signs have their intensity associated with the etiological agent *T. b. rhodesiense* or *T. b. gambiense* [1,3,6]. In the earlier stages of the disease, the parasites reach the draining lymph nodes and the bloodstream, resulting in perivascular cellular infiltration, haemorrhage, oedema, intermittent fever, adenopathies, skin eruptions, and pruritus [7,8]. As the disease progresses to the second stage, also known as the meningoencephalitis stage, parasites cross the blood–brain barrier, invading the central nervous system (CNS) and causing progressive neurological damage [7,9].

During the neurological phase, individuals may present a wide range of CNS disorders, mainly characterized by psychiatric disorders, circadian cycle deregulation and altered sleep patterns, and consequently death [10]. Early neurological symptoms are correlated with widespread meningeal inflammation [11], characterised by infiltration of the cells in the brain, most prominent in white matter (leukoencephalitis) accompanied by a maker of astrocytes and microglia activation [12,13]. Permeability possibly allows trypanosomes to access deep cerebral tissue, interact with the blood–brain barrier, and generate an intense inflammatory reaction, which results in generalised perivascular cuffing, with T helper and B lymphocytes [14].

*T. brucei* parasites were assumed to cross the blood–brain barrier and settle somewhere among the brain cells. However, the brain is a strictly controlled and immune-privileged area that is entirely surrounded by a dense barrier that covers blood vessels [13]. Consequently, it is difficult to relate CNS invasion to parasite–endothelial interactions. It has been reported that *T. b. gambiense* can cause transient decreases in transendothelial electrical resistance and oscillatory increase in intracellular calcium concentration in human brain microvascular endothelial cells, indicating the ability of this parasite to modulate the dynamics and integrity of endothelial cell monolayers during infection [15]. This analysis shows that the bloodstream-form trypanosome proteases might subsequently process or degrade proteins at the intercellular junctions, enabling parasites to cross the barrier among the cells.

Although the entry into the brain by the trypanosome is still controversial, this phenomenon is more likely to be multifactorial, involving the diffusion of signalling substances originating from the host—nitric oxide (NO), tumor necrotic factor (TNF-α), gamma interferon (γ-IFN), and metalloproteinase—such as the host’s matrix metalloproteinases (MMP) or major surface proteins (MSP) from *T. brucei* parasites [11,15,16,17,18]. The last one allows the parasite to survive in the host’s extracellular environment and to develop an efficient evasion mechanism. During chronic infection, the resilience of parasites strongly contributes to the mechanism of antigenic variation [19].

Antigenic variation is a refined mechanism of African trypanosomes adaptation and represents a pivotal protozoan survival strategy against host immune system. This mechanism consists mainly of the expression of variable surface glycoprotein (VSG) and other surface antigens, such as surface zinc metalloproteinases (MSP) and phospholipase-C (PLC). MSP and PLC proteins are anchored to a glycosylphosphatidylinositol (GPI) residue and are responsible for the release of VSG molecules [20,21]. Together with VSG, MSP and PLC proteins activate and induce a generalized state of immunosuppression in the host through the production of cytokines and chemokines, and consequently, the recruitment of inflammatory cells during infection [22].

During an infection with *T. brucei*, the presence of anti-*T. brucei* antibodies—majority anti-VSG antibodies—is important to control the parasite load in the host bloodstream, nevertheless the mechanism of antigenic variation eventually limits the function of the antibodies in protective immunity (Figure 1). In this process, bloodstream parasites alternate the VSG, cyclically perpetuating the presence of *T. brucei* in the host [23]. *T. brucei* parasites contain thousands of genes responsible for the synthesis of thousands of different types of VSG, obstructing the effector action of the host’s immune system, which is mainly mediated by antibodies [24,25].

Pinger and colleagues [26] demonstrated for the first time that trypanosomes require several days to fully replace their VSG coats following a genetic VSG switch and that these parasites are only vulnerable to clearance by early IgM antibodies for a limited time. However, the process to fully evaluate the impact of VSG switching in the context of infection is still unclear. VSG molecules are not suitable antigen targets for vaccine development due to their antigenic variation mechanism, which makes it difficult to develop a protective immune response during the infection as a result of sequentially expressing different VSGs on their surface.

## 2. Current Drug Target Screening for Human African Trypanosomiasis (HAT)

The treatment of sleeping sickness is based on the disease stage and parasite subspecies. Effective treatment of HAT relies on early detection while the patient is still in the first stage of the disease. There are few drugs registered to treat this condition as reviewed by P. Kennedy [7].

First-stage HAT cause by *T. b. gambiense* is treated effectively with intramuscular pentamidine and intravenously suramin for early-stage *T. b. rhodesiense*. There is only one drug effective in second-stage Eastern HAT, melarsoprol and eflornithine to *T. b. gambiense* infection, respectively [27]. Nevertheless, due to the route of drug administration, such injections are exceedingly painful, and there are reports of treatment failures due to resistance of melarsoprol—the only currently effective treatment against *T. b. rhodesiense*. Eflornithine is less toxic than melarsoprol, however it is difficult to administer and more expensive. Drug administration requires rigorous administration schemes through slow intravenous infusion over several days [28].

Nifurtimox is an orally administered drug registered for the treatment of American trypanosomiasis also known as Chagas disease. A combination of orally administered nifurtimox with intravenous eflornithine is used in HAT treatment, resulting in a significant reduction in time and cost [29]. However, the combination therapy is still only effective against *T. b. gambiense*, and still limited by the complex intravenous administration.

Drug repurposing is a strategy that takes existing drugs or classes of compounds and uses these as a starting point for drug discovery [30]. This approach to drug discovery is to include molecules that are known to be active against the intended target or a related target. SCYX7158 and fexinidazole are being evaluated in clinical trials [31,32]. Fexinidazole may be approved as the first all-oral treatment for gambiense HAT and appropriate for both the first and second stage of the disease [33].

The post-genomic age has increased the identification of novel kinds of drug targets for sleeping sickness treatments. A drug target is defined as a protein that is essential for the parasite and does not have homologs in the host [28,30]. Once all proteins that are essential and unique to the parasite are identified, inhibitors may be found by high-throughput screening [30]. Recently, some studies have reviewed potential new pharmacological targets for the development of anti-*T. brucei* drugs. Some of these targets including: cysteine proteases (rhodesain), protein kinase GSK3 (TbGSK3), *N*-myristoyl transferase (TbNMT), glycolytic enzyme phosphofructokinase (PFK), and Methionyl-tRNA synthetase (TbMetRS); important targets to parasite survival [28,30,34]. However, these targets still need to be pharmacologically validated for the development of new drugs.

Here, we discuss the possibility of MSP and PLC proteins being involved in the sophisticated mechanism of antigenic variation, contributing to parasite survival during the maintenance of infection in the mammalian host. This intervention model is mainly based on works published by Grandgenett and colleagues [20] and Gruszynski and colleagues [21], where the authors reported that MSP and PLC are involved in the cellular strategy for removing VSG quickly during bloodstream to procyclic *T. brucei* differentiation. Thus, the aim of this review is to create a reflection about the VSG mechanisms of cleavage and release. Consequently, we intend to explore this mechanism as a potential pathway to develop the much needed new drugs and vaccines against Human African Trypanosomiasis.

## 3. Mechanism of VSG Release as a Suitable Target against African Trypanosomiasis

*T. brucei*, a protozoan that is exclusively extracellular, escapes the host immune system by periodically switching the VSG [35]. The mechanism is mediated by VSG molecules anchored to the cell membrane by glycosylphosphatidylinositol (GPI), which form a physical and biological barrier on the parasite surface making it difficult for the host immune system to have an effective action.

During infection, parasites are continually exposed to the host’s immune system. The presence of antibodies against *T. brucei*, mainly anti-VSG antibodies, is important to control the parasite load in the host bloodstream (Figure 1).

Nevertheless, the mechanism of antigenic variation limits the function of antibodies in protective immunity. The importance of the role of antibodies in protective immunity during *T. brucei* infection can be demonstrated in experimental immunization protocols. Data produced in our laboratory show that DNA vaccines may be a strategy to induce humoral immunity, however, it causes only partial protection when animals are experimentally infected with *T. brucei* [36,37].

Soluble VSG is a strategy used by the parasite for modulating the host’s immune response [20]. VSG molecules are released from the cell surface in two known ways—proteolysis mediated by major surface proteases (MSP) and GPI hydrolysis mediated by phospholipase-C (PLC), both expressed in the bloodstream form of *T. brucei*. This process is presented schematically in Figure 2.

PLC is a phospholipase-C enzyme anchored to a GPI residue that releases a full-length VSG protein from the *T. brucei* surface [33]. Both PLC and MSP proteins are responsible for the mechanism of VSG release from the surface of *T. brucei*. Both enzymes participate synergistically in VSG loss during differentiation from the bloodstream form to the procyclic form of *T. brucei* [24,25]. In this context, MSP and PLC enzymes are involved in the cellular strategy to remove VSG quickly during bloodstream to procyclic differentiation.

Therefore, the mechanism of VSG molecules release could be explored and used with the following research strategies: (i) determination of the enzymatic rate of release of VSG molecules mediated by both MSP and PLC enzymes; (ii) identification of cleavage site based on the VSG sequence domain A and B, available on database [38]; (iii) screening compounds as potential inhibitors for both MSP and PLC enzymes as a strategy for the design of new drugs against *T. brucei*; (iv) the use of MSP and PLC antigens can act as potential targets for vaccine development allowing the formation of antibodies that can interfere and/or block the VSG release mechanism in *T. brucei*.

Although, *T. brucei* and mammalian GPI biosynthetic pathways have much in common, giving rise to the same GPI core structure is significantly different [39], thus providing encouragement for the development of drug leads based on inhibitors of the GPI pathway [39]. There are a few inhibitors already described for PLC and MSP enzymes. For example, *p*- chloromercurylphenylsulphonic and GPI analogues are inhibitors for the PLC enzyme [40,41]. GPI analogues were toxic to parasites, but not to human cells. *T. brucei* infection also causes an increase in the expression of matrix metalloproteinases (MMP). Tetracycline, doxycycline, minocycline, and 1, 10 phenanthroline can inhibit MMP during *T. brucei* infection and inhibit *T. brucei*-MSP [12,14,16,42]. Though information in the field is scarce, inhibiting release enzymes to allow an effective pharmacological and immune system response against the parasite shows promise as a future therapy strategy.

## 4. Major Surface Proteases from *Trypanosoma brucei*: An Important Pathway for the Proteolytic Release of VSG

Major surface proteases (MSP) are present on the surface of Kinetoplastida protozoa *T. brucei*, *Leishmania* spp., and *T. cruzi* [23,24]. MSP proteins are known to play important roles in parasite virulence and in the pathogenesis of infections caused by Kinetoplastida [43]. In *Leishmania* spp., the MSP is called GP63 (63 kDa glycoprotein), a class of zinc metalloproteases abundant on the surface of the parasite, and one of the enzymes best studied and characterized in this parasite [44].

*T. brucei* parasites also encode three homologous proteins of *Leishmania*-GP63, denominated as TbMSP-A, TbMSP-B, and TbMSP-C. According to LaCont and colleagues [24], TbMSP-A and -B classes are located in the C-terminal domain of GP63 and are responsible for the hydrophobic nature of these proteins. On the other hand, TbMSP-C class has a hydrophilic profile.

The elimination of the initial population of *T. brucei* through the production of anti-VSG antibodies induces the mechanism of VSG release through the production of TbMSPs, which are responsible for the cleavage and release of VSG from the surface of the parasite [21]. This process is regulated in the bloodstream by *T. brucei* at the level of translation, localization, or processing of a new class of VSG molecules [24]. So, the maintenance of the infection is assured by the mechanism of antigenic variation and the release of VSG, that causes chronic infection in the host. However, the mechanisms of action and regulation of TbMSPs of *T. brucei* are not fully understood.

Another important consideration regarding the biological role of VSG *T. brucei* is that these parasites are resistant to lysis mediated by the complement system, and this phenomenon is triggered by the dense coating of VSG on the surface of the parasite [21]. Thus, the use of TbMSPs inhibitors could be an interesting strategy to prevent the release of VSG molecules and consequently block the mechanism of antigenic variation, which is important for the chronicity of infection and maintenance of the parasite–host interaction.

As shown schematically in Figure 1, the VSG molecules also function as a physical barrier to protect invariant antigens on the surface of *T. brucei* against the host immune system [45]. Additionally, some studies demonstrated an increased expression of matrix metalloproteinases during infection with *T. brucei* [14,45]. Consequently, through the degradation of the extracellular matrix, the parasite is able to invade some tissues during the infectious process [17]. In this context, the use of metalloproteinase inhibitors during *T. brucei* infection could be an important pharmacological strategy to control the maintenance of the parasite–host interaction.

The catalytic activity of MSP is similar to metalloproteases (MMP) of mammals [46,47] with which it shares several activities, such as extracellular matrix degradation, localization at the cell surface, and zinc-dependent proteolytic activity [48]. It has been shown that the metalloproteases expressed in the *T. brucei* bloodstream display hydrolytic activities on proteins like gelatine, casein, and matrix proteins such as collagen [42].

The highly abundant surface metallopeptidase of *T. brucei* contributes to a myriad of well-established functions in the interaction of this parasite with the mammalian host. The knowledge of biological mechanisms and interaction of MSP expression profile during *T. brucei* infection can be an opportunity for the identification of new therapeutic targets and, consequently, for the development of new pharmacological interventions.

## 5. Contribution of the PLC Enzyme in the Removal of VSG on the Surface of *Trypanosoma brucei*

*T. brucei* also has an enzyme called phospholipase C (PLC) of approximately 40 kDa and is anchored on the surface of the parasite by a GPI residue. PLC does not have a N-terminal signal peptide or a transmembrane domain, so it has the characteristics of an integral membrane protein [49,50,51].

The PLC enzyme is capable of hydrolyzing the GPI anchor present in the VSG molecules on the surface of *T. brucei*, releasing the dimyristyl glycerol component. In this way, PLC participates in the VSG release mechanism [43]. The enzymatic hydrolysis, produced by PLC, results in the conversion of the VSG hydrophobic membrane form (mfVSG) into a water-soluble VSG (sVSG), resulting in the release of the VSG from the parasite membrane. This reaction can be detected by immunoidentification of an epitope contained in the residue of the anchor attached to the VSG, the cross-reacting determinant (CRD) [52,53,54].

GPI–PLC is present only in the metacyclic and bloodstream stages of *T. brucei*. Procyclic trypanosomes do not possess GPI–PLC because, in this stage, GPI anchors contain an extra acylation in the inositol ring [55]. GPI–PLC is unable to release VSG of another parasite when expressed/released by one trypanosome, it only targets its own plasma membrane [56]. Furthermore, VSG is not the only substrate cleaved by GPI–PLC, this protein also acts on other protein substrates of lower and higher molecular mass, like other GPI biosynthetic intermediates [56].

GPI–PLC, besides participating in the mechanism of VSG release, also acts as an important virulence factor during infection with *T. brucei* [45,57]. During the infection, trypanosomes may release pro-inflammatory components, including GPI anchor residues from VSG sheds that trigger NFκB and MAPK, signaling pathways that result in the transcription of pro-inflammatory genes, performing important immunological modulation [54]. The deletion of the PLC gene in *T. brucei* does not eliminate the infectivity of this parasite in animal models. However, during the infection, animals present a lower parasite load and higher survival when compared to the experimental infections caused by wild-type parasites [45].

In general, *T. brucei* metalloprotease is responsible for the degradation of proteins of the blood–brain barrier, such as collagen and laminin, which is possibly pivotal at the transition phase and hemolymphatic phase, to meningoencephalitis [53]. When the expression of the GPI–PLC enzyme in the procyclic forms of *T. brucei* occurs, the release of VSG is ensured by the constant expression of the MSP enzyme [20,44].

## 6. Conclusions and Perspectives

Antigenic variation is an efficient immune evasion mechanism developed by African trypanosomes to survive as extracellular parasites within the mammalian host environment. VSG molecules show extraordinary antigenic diversity, comparative analysis of their protein sequences suggests conserved elements that can be a suitable target for the development of drugs against African trypanosomiasis.

We firstly encourage in this review the importance of future lines of research in the area of enzymatic inhibition of the VSG release mechanism—mediated by MSP and PLC enzymes—as strategies for the development of HAT treatment. Perhaps the major surface proteases (MSPs), PLC crystal structure resolutions, and identification of the proteolytic localization of VSG sites could contribute to the development of new drugs against *T. brucei*. Therefore, the pattern of release VSG molecules of *T. brucei* could be used with the following research strategies: (i) determination of enzymatic rate of release of the VSG molecules mediated by both MSP and PLC enzymes; (ii) screening compounds as potential inhibitors for both MSP and PLC enzymes as a strategy for the design of new drugs against *T. brucei*; and (iii) the use of MSP and PLC proteins as potential antigens for vaccine development, allowing the formation of antibodies that can interfere and/or block the VSG release mechanism in *T. brucei*. These facts made us conclude that the mechanism of VSG release may be an appropriate target for the development of drugs and/or vaccines against African trypanosomiasis.

## Figures and Tables

**Figure 1 ijms-20-01484-f001:**
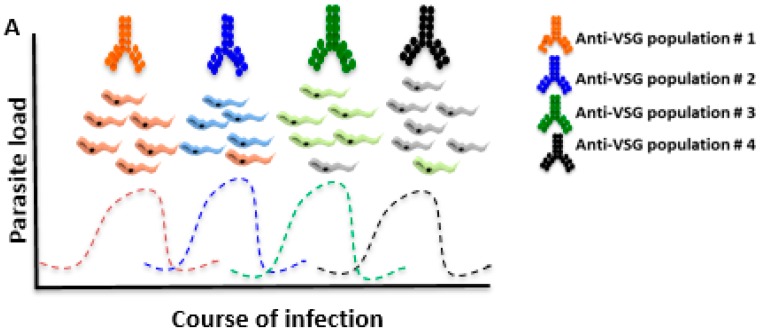
Host immunosuppression caused by coats change of variable surface protein (VSG). *T. brucei* is exclusively an extracellular parasite which constantly changes VSG to successfully evade the mammalian host’s immune system. After antibodies titer increases, the vast majority of parasites are eliminated and only parasites with different VSG coats survive. The bloodstream forms must survive as free-living by assuming a succession of proliferative and quiescent development. The VSG coat is highly antigenic and produces robust VSG-specific antibodies (Anti-VSG population), which participate efficiently in the processes of opsonization and parasite lysis mediated by the complement system. Anti-VSG antibodies contribute to the elimination of parasitic burden, however, a small proportion of the parasite population switches VSG coats, which stimulates a new antibody response to the prevalent new VSG population, and this process repeats until the immune system fails.

**Figure 2 ijms-20-01484-f002:**
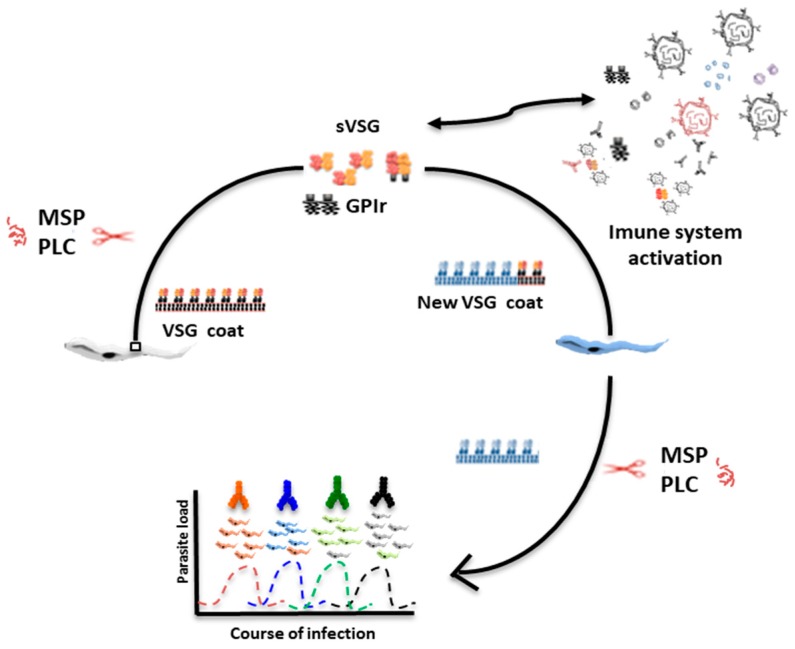
Mechanism of VSG release in *Trypanosome brucei*. Trypanosomes avoid detection by host antibodies by recurrently switching to new VSG coats through major surface proteases (MSP) and phospholipase-C (PLC) cleavage. MSP and PLC enzymes seem to have a synergistic activity in the efficient release of VSG molecules on the surface of the parasite. The trypanosome surface coat is extremely dense, covered by millions of repeats of developmentally specified proteins. The mechanism of VSG cleavage releases soluble VSG (sVSG) and glycosylphosphatidylinositol (GPI) residues (GPIr). Both, sVSG and GPIr are able to activate and modulate immune system response by removal of preformed circulating antibodies and formation of immune complexes. The transitions in composition of the surface coat that covers the plasma membrane is essential for *T. brucei* survival in the host.

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
