# Peer review of "Trypanosoma brucei Interaction with Host: Mechanism of VSG Release as Target for Drug Discovery for African Trypanosomiasis"

_ijms, 2019, doi:10.3390/ijms20061484_

Reviewer 1 Report
This manuscript reviews the mechanism of antigenic variation, the process of VSG release in trypanosoma, and proposed that it may be used as a drug target for trypanosomiasis. It is overall well written, with adequate background and explanations.
Major points:
1. Various of existing drug targets are not reviewed in this manuscript. Authors may consider adding a paragraph to the introduction.
2. Is VSG releasing as a drug target completely new? I cannot find a reference doing the same. If it's indeed novel, authors may want to emphasize it in title, abstract and text.
3. The texts ("Removal", "Modulates" & "Formation") in Figure 1 are placed in a confusing way.
Minor points:
1. Figure 1: PL-C changes to PLC
2. Line 90: "This last one" is confusing.
2. Language/grammar
Line 225: "The pattern of release VSG molecules"
Line 228: "using ... can act as a ..."
Author Response
This manuscript reviews the mechanism of antigenic variation, the process of VSG release in trypanosoma, and proposed that it may be used as a drug target for trypanosomiasis. It is overall well written, with adequate background and explanations.
Major points:
1. Various of existing drug targets are not reviewed in this manuscript. Authors may consider adding a paragraph to the introduction.
In the new version of the manuscript, the introduction session was changed considering the main drugs used during infection with Trypanosoma brucei.
2. Is VSG releasing as a drug target completely new? I cannot find a reference doing the same. If it's indeed novel, authors may want to emphasize it in title, abstract and text.
Yes. To date, no paper suggests the VSG release model as a strategic target for the design of drugs and vaccines. In the manuscript, the mechanisms of VSG release are discussed based initially on the work of Grandgenett and colleagues, where the authors reported that MSP and PLC are involved in the cellular strategy for removing VSG quickly during bloodstream to procyclic T. brucei differentiation. The manuscript reflects upon the use of this mechanism as a potential target for the development of drugs and vaccines, since the mechanism of antigenic variation limits the development of vaccines using VSG as an antigenic candidate.
3. The texts ("Removal", "Modulates" & "Formation") in Figure 1 are placed in a confusing way.
From figure 1 two new figures were elaborated (figure 1 and figure 2).
Minor points:
All proposed minor reviewers were updated in the new version of the manuscript.
1. Figure 1: PL-C changes to PLC
2. Line 90: "This last one" is confusing.
2. Language/grammar
Line 225: "The pattern of release VSG molecules"
Line 228: "using ... can act as a ..."
Reviewer 2 Report
The manuscript by Moreno and co-workers is a somewhat interesting read, but I don’t think it should be accepted for publication as a review article because of these reasons:
1. Much of the information presented in the manuscript are well-established concepts in trypanosomatid biology.
2. Most of the cited papers are dated and are not comprehensive enough to warrant review.
Of course, the reason is probably because that there is relatively few basic research on protozoan organisms in general.
3. The drug targets suggested by the authors have not be validated in such a way that resources should be devoted to pursuing drug discovery (which is the central thesis of the review).
My recommendation is that the authors should consider rewriting this as a perspective. Perhaps, by reducing much of the introduction and providing some suggestions on what should be done to fully validate the suggested targets.
Author Response
Reviewer #2:
The manuscript by Moreno and co-workers is a somewhat interesting read, but I don’t think it should be accepted for publication as a review article because of these reasons:
1. Much of the information presented in the manuscript are well-established concepts in trypanosomatid biology.
The central objective of the manuscript is to bring a discussion and reflection about the mechanism of antigenic variation, a well-established biological concept in T. brucei. However, it was emphasized the mechanism of the release of VSG, an important step for the phenomenon of antigenic variation. There is no literature review regarding the mechanism of VSG release, but rather the mechanism of antigenic variation.
Is the mechanism of VSG release a model to be considered for the development of drugs and vaccines against T. brucei? This is the reflection that the authors intend to induce in the readers.
2. Most of the cited papers are dated and are not comprehensive enough to warrant review.
Of course, the reason is probably because that there is relatively few basic research on protozoan organisms in general.
The papers referred to in the manuscript are the most relevant in the context of the VSG release mechanism in T. brucei. Therefore, it emphasizes the scarcity of literature in the field and therefore the need for more publications in the area.
3. The drug targets suggested by the authors have not be validated in such a way that resources should be devoted to pursuing drug discovery (which is the central thesis of the review).
The drugs discussed in the review were not validated against the VSG release model. In this context, the authors intend to provoke future lines of research in this area, mainly because it is a neglected tropical disease.
My recommendation is that the authors should consider rewriting this as a perspective. Perhaps, by reducing much of the introduction and providing some suggestions on what should be done to fully validate the suggested targets.
The manuscript was rewritten, mainly the introductory session, as well as the addition of information regarding the therapy used in human infection with T. brucei. The authors believe that the manuscript will contribute to the validation of VSG release model as a further pharmacological/immunological targeting option for future research in drug discovery and vaccination against T. brucei.
Reviewer 3 Report
The review by Moreno et al., suggests exploring the mechanisms of antigenic variation of T. brucei as a potential target for the development of new trypanosomiasis drugs.
The topic may be potentially interesting, however additional details at the molecular level have to be added.
In particular:
- introduce a more detailed description of the current strategy to treat these pathologies and clarify the importance to discover new targets for anti-T. brucei drugs
- clarify the sentence page 4 lines 154-158 regarding the structure of MPS of T. brucei
- clarify the sentence page 5 lines 191-192. The authors describe PLC as a protein without a transmembrane segment but then write “it has the characteristic of an integral membrane protein”
- the figure appears confused.
Author Response
Reviewer #3:
The review by Moreno et al., suggests exploring the mechanisms of antigenic variation of T. brucei as a potential target for the development of new trypanosomiasis drugs.
The topic may be potentially interesting, however additional details at the molecular level have to be added.
In particular:
1. Introduce a more detailed description of the current strategy to treat these pathologies and clarify the importance to discover new targets for anti-T. brucei drugs
In the new version of the manuscript the session introduction was restructured with addition of the drugs used in the therapy Human African Trypanosomiasis (HAT – T. brucei). A new session was introduced to the manuscript: “current drug target screening for Human African Trypanosomiasis – HAT”.
2. Clarify the sentence page 4 lines 154-158 regarding the structure of MPS of T. brucei.
The sentence was rewritten in the new version of the manuscript.
3. Clarify the sentence page 5 lines 191-192. The authors describe PLC as a protein without a transmembrane segment but then write “it has the characteristic of an integral membrane protein”
The sentence was rewritten in the new version of the manuscript.
4. The figure appears confused.
Figure 1 was segmented into two new figures (figure 1 and figure 2).
Round 2
Reviewer 2 Report
The manuscript has improved, although the review can be a bit better.
The prose is somewhat convoluted in many parts, but the science in fine. I have highlighted parts of the manuscript that should be edited.
The format of the references should be consistent.

Author Response
The answers to the reviewer (point by point):
1. The manuscript has improved, although the review can be a bit better.
The new version of the manuscript presents new changes and revisions, all highlighted in the text in track changes.
2. The prose is somewhat convoluted in many parts, but the science in fine. I have highlighted parts of the manuscript that should be edited.
All parts of the manuscript highlighted by the reviewer were edited in the new version of the manuscript. The new file displays all the changes highlighted.
3. The format of the references should be consistent.
The session "references" was all adapted equally, according to the suggestions of the reviewer.
Reviewer 3 Report
The authors answered the requests
Author Response
According to the reviewer, all questions were answered after round 1. However, a more detailed review of English was made and highlighted in the new version of the manuscript.